# Generative Human Geometry Distribution

**Xiangjun Tang**
KAUST
xiangjun.tang@kaust.edu.sa

**Biao Zhang**[*]
KAUST
biao.z@outlook.com

**Peter Wonka**
KAUST
pwonka@gmail.com

## Abstract

Realistic human geometry generation is an important yet challenging task, requiring both the preservation of fine clothing details and the accurate modeling of clothing-body interactions. To tackle this challenge, we build upon Geometry distributions—a recently proposed representation that can model a single human geometry with high fidelity using a flow matching model. However, extending a single-geometry distribution to a dataset is non-trivial and inefficient for large-scale learning. To address this, we propose a new geometry distribution model by two key techniques: (1) encoding distributions as 2D feature maps rather than network parameters, and (2) using SMPL models as the domain instead of Gaussian and refining the associated flow velocity field. We then design a generative framework adopting a two-staged training paradigm analogous to state-of-the-art image and 3D generative models. In the first stage, we compress geometry distributions into a latent space using a diffusion flow model; the second stage trains another flow model on this latent space. We validate our approach on two key tasks: pose-conditioned random avatar generation and avatar-consistent novel pose synthesis. Experimental results demonstrate that our method outperforms existing state-of-the-art methods, achieving a $57\%$ improvement in geometry quality.

## 1 Introduction

3D human geometry generation is a critical yet challenging task. The human body exhibits high-frequency clothing details, which are inherently difficult to synthesize. Therefore, the core of this task is the design of a representation that can capture both the underlying structure and fine details, posing two key challenges: 1) encoding high-frequency details into a low-dimensional manifold without losing fidelity; 2) modeling the relationship between clothing wrinkles and poses to preserve realistic details.

Existing methods represent the human body in different ways. NeRFs (Men et al., 2024; Zhang et al., 2022; Xiong et al., 2023; Jiang et al., 2023; Xu et al., 2023; Zheng et al., 2024) focus on the rendering results but neglect underlying geometry and are constrained by the rendering speed and resolution. Implicit functions (Xu et al., 2022), such as signed distance functions, struggle to synthesize thin structures and tend

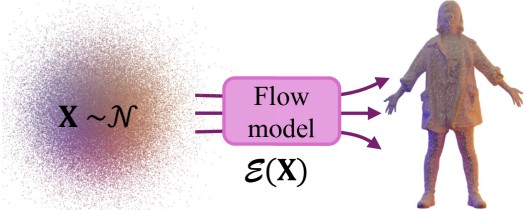

Figure 1: Geometry distribution.

to oversmooth results. Point clouds (Zhang et al., 2024c;d) and volume-based representations trade off between memory efficiency and quality. Recently, geometry distributions (Zhang et al., 2025) model single 3D shapes as distributions of points on their surfaces. Samples $\mathbf{X}$ from a Gaussian distribution $\mathcal{N}$ are transformed via a flow-based diffusion network $\mathcal{E}(\mathbf{X})$ into the target geometry, capturing fine-grained surface details (see Fig. 1). This formulation enables infinite point sampling, allowing for a high-fidelity representation of individual shapes. However, extending a single-geometry distribution to an entire dataset is non-trivial because of two reasons: 1) single-geometry formulations store the geometry in the parameters of a flow network, which leads to prohibitive memory consumption and limits their scalability for generative tasks, and 2) while learning the flow velocity fields from Gaussian distributions to a single shape is feasible, extending this to multiple

---

[*]Corresponding Author

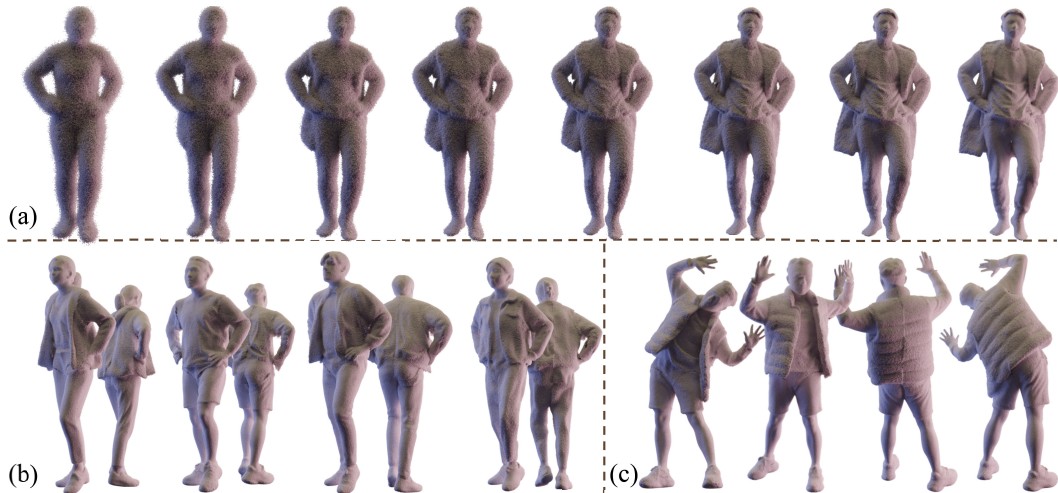

Figure 2: (a) Denoising process of a human geometry distribution. (b) Random avatar generation for a given pose. (c) Novel pose generation of a given avatar. Results are rendered from point clouds.

shapes across a dataset becomes computationally expensive and inefficient. To this end, we propose a novel human geometry distributions model by two key techniques:

- We encode each human geometry distribution into a feature map rather than the network weights of a flow network directly, providing a generalized way to represent geometry distributions.

- We adopt the SMPL (Pavlakos et al., 2019) model distribution instead of a Gaussian and refine the associated flow velocity field for efficient learning. Specifically, we construct spatially efficient mappings from the velocity field and reformulate them into a regularized dense space.

Based on these designs, we propose the first 3D generative method for geometry distributions, modeling the distribution of individual human geometry distributions. Our framework is formulated in two stages: first, we use a flow model to compress each human geometry distribution into a compact feature map, from which a high-fidelity human geometry can be sampled through a denoising process (Fig. 2 (a)), and then, we train another flow model to generate human geometry distributions (i.e., the feature maps). While the currently top-performing methods (Xu et al., 2022; Zhang et al., 2024d; 2023b) obtain visually plausible geometric details by enhanced rendering techniques, our method synthesizes high-fidelity geometry directly, leading to large improvements. We design two generative tasks to evaluate the effectiveness of our method (Fig. 2 (b) and (c)). The first task generates random 3D human geometries conditioned on a given pose, while the second enables novel pose generation of a given avatar. Our quantitative results show that our method improves the geometry quality by 57% (42.9 to 16.2) and the visual appearance by 7% (17.4 to 16.2) compared to the state-of-the-art.

## 2 RELATED WORK

### 2.1 GENERAL 3D GENERATION

3D generation methods employ various representations, such as meshes, voxels, point clouds, and implicit functions, each offering distinct advantages but also facing inherent limitations (Zhang et al., 2025). Here, we focus on compact representations that scale efficiently to large datasets.

Given one or more images, direct inference of pixel-aligned positions (Yu et al., 2021a; Tang et al., 2024; Xu et al., 2024b;a) avoids intermediate representations, which is unsuitable for modeling intrinsic human shapes. A popular alternative is to synthesize tri-planes as intermediate representations (Wang et al., 2023; Li et al., 2023; Hong et al., 2023), from which implicit functions or points (Zou et al., 2024; Han et al., 2024) can be sampled. However, tri-planes are limited by their resolution, restricting their ability to capture fine-grained details. GaussianCube (Zhang et al., 2024a) transfers Gaussians into a regular voxel grid via optimal transport, limiting its flexibility for

Table 1: Comparison of 3D human representation methods.

| Method | Loose Clothing | Scalability | Fine Details |
|---|---|---|---|
| Tri-plane + NeRF (Noguchi et al., 2022; Zhang et al., 2022) | ✓ | ✓ | ✗ |
| Template-based (Hong et al., 2022; Hu et al., 2024) | ✗ | ✓ | ✗ |
| Primitive Volumes (Chen et al., 2023) | ✓ | ✓ | ✗ |
| Mesh-based (Sanyal et al., 2024; Xu et al., 2024c; Feng et al., 2023) | ✗ | ✓ | ✗ |
| Point-based (Abdal et al., 2024; Zhang et al., 2024d) | ✓ | ✓ | ✗ |
| Implicit Function (Xu et al., 2022; Zhang et al., 2023b) | ✓ | ✓ | ✗ |
| Geometry Distributions (**Ours**) | ✓ | ✓ | ✓ |

intricate articulation. Gamba (Shen et al., 2024) and VectorSet (Zhang et al., 2023a) employ cross-attention mechanisms to inject 3D embeddings or image tokens, albeit constrained by memory consumption. Trellis (Xiang et al., 2024) leverages sparse voxels but lacks mechanisms to incorporate human-specific priors. In contrast, 2D maps (Rai et al., 2025; Elizarov et al., 2024; Yan et al., 2024), offer a memory-efficient alternative that aligns naturally with human priors. Another line of work learns a "distribution-of-distributions" (Yang et al., 2019; Cai et al., 2020); while conceptually similar, these methods are limited to coarse shapes. Effectively learning a distribution of high-fidelity geometry distributions remains an open challenge.

## 2.2 3D HUMAN RECONSTRUCTION AND GENERATION

Features extracted from images (Saito et al., 2019; 2020; Alldieck et al., 2022) provide rich view-dependent information, enabling high-fidelity human reconstruction. Some works incorporate additional cues such as voxel grids (He et al., 2020), human templates (Zheng et al., 2021), or multi-view images (Shao et al., 2022) to mitigate the depth ambiguity of single-view image. For example, ICON (Xiu et al., 2022) and ECON (Xiu et al., 2023) predict the frontal and back normals from the SMPL template and reconstruct the human using these maps. Building on these reconstruction techniques, generative networks—such as GANs (Xiong et al., 2023; Jiang et al., 2023) and diffusion models (Sengupta et al., 2024; Shao et al., 2022; Zhang et al., 2024b)—have been employed to synthesize pixel-aligned features or generate multi-view images (Kolotouros et al., 2024) for human generation. However, these methods are inherently limited by their reliance on view-dependent features, sensitivity to multi-view inconsistencies, and often require post-processing to fuse information from different views (Zhang et al., 2024b), failing to capture the intrinsic 3D shape.

To learn intrinsic 3D representations, a popular approach (Noguchi et al., 2022; Zhang et al., 2022; Yang et al., 2024; Wu et al., 2023; 2024) is to train a GAN model that synthesizes tri-planes and uses NeRF for rendering. However, these methods are constrained by rendering speed and resolution, often requiring super-resolution modules (Bergman et al., 2022; Dong et al., 2023), multi-part structures (Xu et al., 2023), or refinement stages (Men et al., 2024; Zheng et al., 2024) to achieve high-fidelity results. In addition, Gaussian3Diff (Lan et al., 2024) employs a diffusion model to learn the tri-plane representation for 3D heads, providing an alternative to GAN-based approaches. Rather than employing tri-planes, alternative methods incorporate human templates, alleviating sparse spatial occupancy, but struggle with loose clothing (Hong et al., 2022) or still rely on super-resolution modules (Hu et al., 2024). Explicit representations, such as primitive volumes (Chen et al., 2023), provide efficient rendering but face challenges in extracting detailed geometry. Mesh-based methods, using displacement (Sanyal et al., 2024) or layered surface volumes (Xu et al., 2024c), are limited by fidelity and struggle with loose clothing. Point-based representations (Abdal et al., 2024; Zhang et al., 2024d;c) offer flexibility for modeling thin structures but are constrained by point density. Furthermore, it can be difficult to produce accurate 3D geometry under indirect supervision from rendering. In contrast, gDNA (Xu et al., 2022) proposes learning implicit functions directly from 3D data, but synthesizing high-fidelity geometry remains challenging due to inherent representation limitations (Zhang et al., 2025). Our approach directly learns from 3D data while enabling "infinite" sampling, leading to high-fidelity geometry synthesis. See Tab. 1 for comparison.

## 3 PRELIMINARIES

**Flow Matching** Flow matching (Lipman et al., 2023) is a variant of diffusion models. It constructs a flow that transforms samples from a source distribution $\mathbf{p}$ to a target distribution $\mathbf{q}$. Specifically, the training objective loss is defined as follow:

$$\arg\min_{\theta} \mathbb{E}_{\mathbf{x}_0 \sim \mathbf{p}, \mathbf{x}_1 \sim \mathbf{q}, t \in [0,1]} \|u_\theta(\mathbf{x}_t, t) - (\mathbf{x}_1 - \mathbf{x}_0)\| \tag{1}$$

where $\mathbf{x}_t = (1 - t)\mathbf{x}_0 + \mathbf{x}_1$. After training, an ordinary differential equation is solved to transition from $\mathbf{x}_0$ to $\mathbf{x}_1$, allowing us to sample $\mathbf{x}_1 \sim \mathbf{q}$ by first sampling $\mathbf{x}_0 \sim \mathbf{p}$.

Based on the flow matching formulation, the core of our modeling is to identify a suitable source distribution and a target distribution that aligns well with our problem. For brevity, we omit $t \in [0, 1]$ in subsequent equations.

**Geometry Distributions**   Geometry distributions model a surface $\mathcal{M} \subset \mathbb{R}^3$ as a probability distribution $\Phi_{\mathcal{M}}$, where any sample $\mathbf{x}_1 \sim \Phi_{\mathcal{M}}$ corresponds to a point on the surface. To achieve this, Zhang et al. (2025) adapt a diffusion model to learn the transformation from a Gaussian distribution $\mathcal{N}(\mathbf{0}, \mathbf{1})$ to the target distribution of surface points. A point $\mathbf{x}_1 \in \mathcal{M}$ on the target surface can then be obtained by solving an ODE with an initial point $\mathbf{x}_0 \sim \mathcal{N}(\mathbf{0}, \mathbf{1})$.

## 4 METHOD

### 4.1 OVERVIEW

In this section, we first introduce our human geometry distribution formulation by identifying suitable source and target distributions (Sec. 4.2). Next, we describe encoding this distribution into a feature map using an auto-decoder architecture, as depicted in Fig. 4 (a) (Sec. 4.3). Finally, we present generative tasks aimed at synthesizing the feature map, guided by the SMPL template, optionally incorporating additional conditioning inputs, as illustrated in Fig. 4 (b) (Sec. 4.4).

### 4.2 HUMAN GEOMETRY DISTRIBUTION

Our insight is to replace the Gaussian distribution $\mathcal{N}(0, 1)$ in the prior geometry distributions (Zhang et al., 2025) with the SMPL template shape distribution $\Phi_{\mathcal{S}}$, aligning the source distribution closer to the target geometry distribution $\Phi_{\mathcal{T}}$. A naive approach can be formulated as:

SMPL templates                    Human geometries

$$\arg \min_{\theta} \mathbb{E}_{\mathbf{x}_0 \sim \Phi_{\mathcal{S}}, \mathbf{x}_1 \sim \Phi_{\mathcal{T}}} \|u_{\theta}(\mathbf{x}_t, t) - (\mathbf{x}_1 - \mathbf{x}_0)\|, \quad (2)$$

Figure 3: Aggregated samples $(\mathbf{x}'_0, \mathbf{x}_1)$.

where $\mathbf{x}_t = (1 - t)\mathbf{x}_0 + \mathbf{x}_1$. Building upon this, we propose two strategies to reduce the modeling complexity and improve training efficiency, including explicitly constructing training pairs and normalizing samples from distributions.

**Training Pair Construction.**   Since the learned probability flow by Eq. 2 approximates a conditional optimal transport (Lipman et al., 2024), one strategy for improving training efficiency is avoiding learning extraneous paths between distant points. To achieve this, we prioritize short flow by constructing training pairs set $\{(\mathbf{x}'_0, \mathbf{x}_1)\}_{\mathcal{T}}$ where $\mathbf{x}'_0 \sim \Phi_{\mathcal{S}}$ is geometrically close to $\mathbf{x}_1 \sim \Phi_{\mathcal{T}}$. Specifically, we first sample a set of points $\{\mathbf{x}_0\}_{\mathcal{S}}$ on the SMPL template and a set of points $\{\mathbf{x}_1\}_{\mathcal{T}}$ on the target geometry. Then, for each $\mathbf{x}_1 \in \{\mathbf{x}_1\}_{\mathcal{T}}$, the corresponding $\mathbf{x}'_0$ is obtained by:

$$\mathbf{x}'_0 = \arg \min_{\mathbf{x}_0 \in \{\mathbf{x}_0\}_{\mathcal{S}}} \|\mathbf{x}_1 - \mathbf{x}_0\|_2.$$

Notably, because multiple points $\mathbf{x}_1$ may share the same nearest SMPL points $\mathbf{x}'_0$, especially in loose or wrinkled regions, directly learning a path from $\mathbf{x}'_0$ to $\mathbf{x}_1$ leads to under-sampled geometry (similar to the artifacts shown in the second row of Fig. 6). Therefore, we add a perturbation drawn from $\mathcal{N}(0, \boldsymbol{\sigma})$ to $\mathbf{x}'_0$, introducing randomness to enhance sample diversity. In this formulation, the source distribution becomes $\mathcal{N}(\mathbf{x}'_0, \boldsymbol{\sigma})$ and the target distribution is the human geometry. Sampling from the constructed training pairs, the flow matching optimization objective is denoted by:

$$\arg \min_{\theta} \mathbb{E}_{\mathbf{x}_0 \sim \mathcal{N}(\mathbf{x}'_0, \boldsymbol{\sigma}), (\mathbf{x}'_0, \mathbf{x}_1) \in \{(\mathbf{x}'_0, \mathbf{x}_1)\}_{\mathcal{T}}} \|\cdot\| \quad (3)$$

**Distribution Normalization.**   When training the transformation from the SMPL template to the human geometry of a dataset (Eq. 3), the network receives spatially imbalanced supervision due to unevenly sampling points, which arises from two factors. First, points lie only on the surfaces, making them extremely sparse and unevenly distributed relative to the full 3D space. Second, variations in human pose and body shape further exacerbate this spatial imbalance, as some regions are more

Figure 4: **Overview of our method.** (a) We encode a geometry into a feature map, which is decompressed with a SMPL vertex map. The decompressed feature serves as a condition for our denoising network. (b) The human generation task is formulated as the conditional generation of feature maps, guided by the SMPL vertex map, optionally incorporating additional conditioning inputs.

dynamic and thus sampled less consistently across geometries. Fig. 3 visualizes this effect by aggregating sampled points from multiple training iterations, showing that some spatial regions have high point density while others remain sparsely sampled. This spatial imbalance leads to inefficient training, as the model pays more attention to high-density regions while neglecting low-density regions. To address this, we normalize both the source and target distribution by subtracting $\mathbf{x}_0'$. The source distribution becomes a zero-centered Gaussian distribution $\mathcal{N}(0, 1)$, with $\boldsymbol{\sigma}$ set to 1, and the target distribution is modeled as a regularized dense displacement field $\Delta\mathbf{x} = \mathbf{x}_1 - \mathbf{x}_0'$. Note that this subtraction removes the positional information of $\mathbf{x}_0'$. To retain this information without suffering from imbalanced sampling, we reintroduce $\mathbf{x}_0'$ as a conditional signal that scales the network hidden feature, indirectly influencing the feature representation. By building flow between regularized dense spaces, we reduce the modeling complexity and achieve improved training efficiency, as demonstrated in Sec 5.2. The optimization objective is denoted by:

$$\arg\min_{\theta} \mathbb{E}_{\mathbf{n}, (\mathbf{x}_0', \mathbf{x}_1)} \left\| u_\theta(\mathbf{x}_t, t | \mathbf{x}_0') - (\Delta\mathbf{x} - \mathbf{n}) \right\|, \tag{4}$$

where $\mathbf{n} \sim \mathcal{N}(0, 1)$, $(\mathbf{x}_0', \mathbf{x}_1) \in \{(\mathbf{x}_0', \mathbf{x}_1)\}_{\mathcal{T}}$ and $\mathbf{x}_t = (1 - t)\mathbf{n} + t\Delta\mathbf{x}$.

## 4.3 CONDITIONAL DISTRIBUTION ENCODING

The dataset for our task consists of a set of pairs $\mathcal{D} = \{(\mathcal{S}, \mathcal{T})\}$, each associated with a latent representation $\mathbf{z}_{\mathcal{T}|\mathcal{S}}$, which is used to recover $\mathcal{T}$ (target geometry) by conditioning on $\mathcal{S}$ (SMPL). This design aligns with our generative task, i.e. , $\mathcal{S}$-conditioned $\mathcal{T}$ generation. The training loss can be written as,

$$\arg\min_{\theta, \{\mathbf{z}_{\mathcal{T}|\mathcal{S}}\}} \mathbb{E}_{(\mathcal{S}, \mathcal{T}) \in \mathcal{D}} \mathbb{E}_{\mathbf{n}, (\mathbf{x}_0', \mathbf{x}_1)} \left\| u_\theta(\mathbf{x}_t, t | \mathbf{x}_0', \mathbf{z}_{\mathcal{T}|\mathcal{S}}) - (\Delta\mathbf{x} - \mathbf{n}) \right\|, \tag{5}$$

To enhance the expressiveness, we model $\mathbf{z}_{\mathcal{T}|\mathcal{S}} \in \mathbb{R}^{C \times H \times W}$ as a compressed 2D feature map (height $H$ and width $W$), as shown in Fig. 4 (a). The representation aligns with recent advancements in UV representations for 3D objects (Yan et al., 2024). Furthermore, we use a decoder network $\mathrm{Dec}_\phi(\cdot)$ to decompress $\mathbf{z}_{\mathcal{T}|\mathcal{S}}$ to a higher resolution. By leveraging the UV coordinates associated with $\mathbf{x}_0'$, we can then obtain per-point latents on the surface $\mathcal{S}$ by sampling the high-resolution map, denoted as $\mathrm{Dec}_\phi(\mathbf{z}_{\mathcal{T}|\mathcal{S}})(\mathbf{x}_0')$. Thus, the revised optimization can be written as

$$\arg\min_{\theta, \phi, \{\mathbf{z}_{\mathcal{T}|\mathcal{S}}\}} \mathbb{E}_{(\mathcal{S}, \mathcal{T}) \in \mathcal{D}} \mathbb{E}_{\mathbf{n}, (\mathbf{x}_0', \mathbf{x}_1)} \left\| u_\theta\left(\mathbf{x}_t, t | \mathbf{x}_0', \mathrm{Dec}_\phi(\mathbf{z}_{\mathcal{T}|\mathcal{S}})(\mathbf{x}_0')\right) - (\Delta\mathbf{x} - \mathbf{n}) \right\|. \tag{6}$$

**Decoder.** As illustrated in Fig. 4 (a), the network $\mathrm{Dec}_\phi(\cdot)$ is a UNet-style network that contains several downsampling and upsampling layers. We render SMPL vertex positions into UV maps and concatenate them with hidden features in the convolution layers.

**Denoising network.** The design of the denoiser $u_\theta$ is adapted from (Zhang et al., 2025). In this case, we have two additional conditioning signals, SMPL point $\mathbf{x}_0'$ and latent $\mathrm{Dec}_\phi(\mathbf{z}_{\mathcal{T}|\mathcal{S}})(\mathbf{x}_0')$. Additionally, we augment $\mathbf{x}_0'$ by concatenating normals and canonical coordinates. Surface normals provide directional cues for clothing inference, while canonical coordinates encode body part semantics (e.g., distinguishing limbs from the torso).

## 4.4 GENERATIVE TASKS

After finishing learning the latents $\{\mathbf{z}_{\mathcal{T}|\mathcal{S}}\}$, we train generative models in the latent space. We investigate two tasks: pose-conditioned random generation and novel pose generation of a given avatar, both built on a U-Net (Karras et al., 2024; 2025) architecture, as illustrated in Fig. 4 (b).

Pose-conditioned random generation synthesizes diverse human geometries conditioned on an SMPL template mesh. To encode pose information, we render SMPL vertex positions into UV maps and inject them as residual connections into the U-Net architecture. For novel pose generation, the model additionally takes a frontal normal image as an extra condition to indicate the avatar identity. Specifically, for each animation sequence in the dataset, we randomly select one frame to provide the normal image condition and another frame to provide the SMPL pose condition. We leverage the frozen DINO-ViT model (Caron et al., 2021) to extract image features, which are then fused into the U-Net via cross-attention layers.

The pose-conditioned generation networks are trained on the THuman2 dataset (Yu et al., 2021b), while the novel pose generation is trained on 4DDress (Wang et al., 2023). See the Appendix for implementation details.

## 5 EXPERIMENTS AND RESULTS

In this section, we first conduct ablation studies on the training pair construction strategy and network architecture. Next, we compare different formulations of geometry distribution, including the formulation proposed by Zhang et al. (2025) and our proposed alternatives. Finally, we compare our method with state-of-the-art generative methods regarding the two selected generative tasks.

### 5.1 ABLATION STUDY

#### 5.1.1 TRAINING PAIRS CONSTRUCTION

Rather than searching for the nearest points on the dense SMPL mesh, we first sample a relatively sparse set of points from the SMPL mesh and then determine the nearest points $\mathbf{x}_0'$ from this sampled set. It distributes the mapping workload across different $\mathbf{x}_0'$, which improves sampling efficiency. To validate this, we train our auto-decoder under both strategies and present the results in Fig. 6.

The first row shows the results using the sparse points set sampling strategy, while the second row corresponds to directly identifying the nearest points on the SMPL mesh. The normal maps in the second row exhibit noticeable holes, indicating insufficient sampling. These artifacts are particularly pronounced for loose clothing (middle case in Fig. 6) but also appear in regions of relatively tight clothing (left and right case in Fig. 6).

#### 5.1.2 NETWORK ARCHITECTURE

We experiment to validate that the auto-decoder outperforms the existing auto-encoders for our task. The auto-encoder learns features from the input, requiring more computations and being potentially constrained by the input's representation capacity. We adopt the 3D representation network VecSet (Zhang et al., 2023a) as the encoder for the auto-encoder and take $50,000$ input points, balancing representation capability and memory consumption. Since VecSet uses

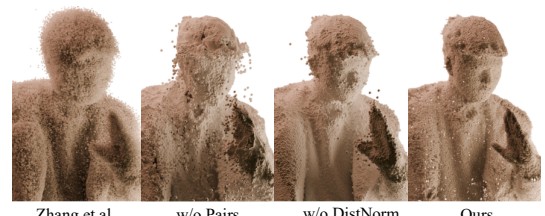

Zhang et al.    w/o Pairs    w/o DistNorm    Ours

Figure 5: Single-geometry fitting visualization.

independent embeddings rather than a feature map, we treat each pixel of the feature map $\mathbb{R}^{24 \times 24 \times 8}$ as an individual embedding, yielding 576 embeddings (FeatureMap). To further ensure a fair comparison, we introduce an additional model where we replace our feature map decompression module (Fig. 4 (b)) with the one proposed by VecSet, maintaining as much of the VecSet architecture as possible (VecSet).

The experiment is conducted on the THuman2 dataset (Yu et al., 2021b). As shown in Tab. 3, we report the average distance between the synthesized points and the surface. While converting independent embeddings to a feature map improves reconstruction accuracy, the results remain inferior to our proposed auto-decoder.

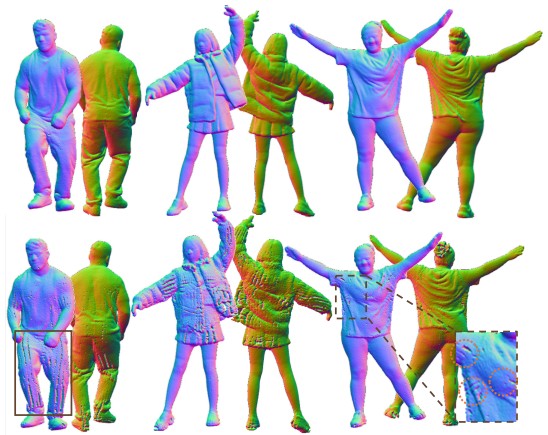

Figure 6: Comparison of strategies for constructing training pairs. The first row shows our proposed strategy, the second row finds the nearest points on the SMPL mesh.

Figure 7: Visualization results of different geometry distribution formulations at the $30,000$th training iteration. GT represents the ground-truth result.

## 5.2 COMPARISONS OF GEOMETRY DISTRIBUTION FORMULATIONS

To evaluate the training efficiency of different geometry distributions, we first conduct a single-geometry fitting task and then extend our analysis to a dataset. We compare our method with several alternatives, including the geometry distribution proposed by Zhang et al. (2025), the naive formulation in Eq. 2 (w/o Pairs), and the formulation sampling from $\mathcal{N}(\mathbf{x}_0', \boldsymbol{\sigma})$ without distribution normalization (w/o DistNorm), proposed in Eq. 3.

For the single-geometry fitting task, we train each formulation for $10,000$ iterations and visualize the synthesized points at the final step in Fig. 5. Our method produces the smoothest and most accurate reconstructions, while all other methods generate points that deviate from the surface. The Chamfer distances at the final step are presented in Tab. 2 (Single). The distribution proposed by Zhang et al. (2025) performs the worst because it samples from Gaussian noise and requires more iterations to capture the coarse human shape. The "w/o Pairs" formulation also exhibits suboptimal performance as it must learn mappings between distant points, leading to slower convergence. The approach that samples from $\mathcal{N}(\mathbf{x}_0', \boldsymbol{\sigma})$ (w/o DistNorm) achieves the lowest Chamfer distance in this task. With a fixed pose, each sample can focus on learning specified local features, thereby simplifying the optimization. However, as shown in Fig. 5, its reliance on dispersed Gaussian centers leads to noisy output for some regions. Furthermore, this approach negatively impacts convergence when scaling to datasets with diverse poses, as reflected in its degraded performance in Tab. 2 (Dataset).

For dataset-scale evaluation, we train each model on the THuman2 (Yu et al., 2021b) dataset for $30,000$ iterations. Our method consistently achieves significantly higher fidelity compared to the other approaches within the same number of training iterations, as shown in Fig. 7. Notably, the results do not represent the final outcome, as training is not fully completed. The evolution of Chamfer distance over iterations is provided in the Appendix.

Table 2: Chamfer distance of different distribution formulations.

|  | Zhang et al. | w/o Pairs | w/o DistNorm | Ours |
|---|---|---|---|---|
| Single | 0.0083 | 0.0040 | **0.0020** | 0.0032 |
| Dataset | 0.0101 | 0.0706 | 0.0071 | **0.0032** |

Table 3: Surface distance comparison between various designs.

| Model | Surface Distance ↓ |
|---|---|
| VecSet | 0.0018 |
| FeatureMap | 0.0014 |
| Ours | **0.0012** |

## 5.3 POSE-CONDITIONED RANDOM GENERATION

This experiment measures the geometry quality of pose-conditioned random generation. Consistent with previous work (Zhang et al., 2023b; 2024d), we utilize the THuman2 Dataset (Yu et al., 2021b) for training and evaluation. We calculate the Fréchet Inception Distance (FID) metrics between the rendered normal images and ground truth normal maps. Notably, some works (Xu et al., 2022; Zhang et al., 2023b; 2024d) enhance their rendering results using information such as normal

Table 4: Comparison of FID scores. The * results are adopted from E3Gen (Zhang et al., 2024d). For some methods, the results are rendered directly from their raw geometries, so the numbers in both rows are identical.

|  | ENARF* | GNARF* | EVA3D* | E3Gen | GetAvatar | gDNA | Ours |
|---|---|---|---|---|---|---|---|
| Raw Geometry | 223.72 | 166.62 | 60.37 | 65.32 | 56.07 | 42.90 | **16.16** |
| Enhanced Rendering | 223.72 | 166.62 | 60.37 | 28.12 | 22.77 | 17.43 | **16.16** |

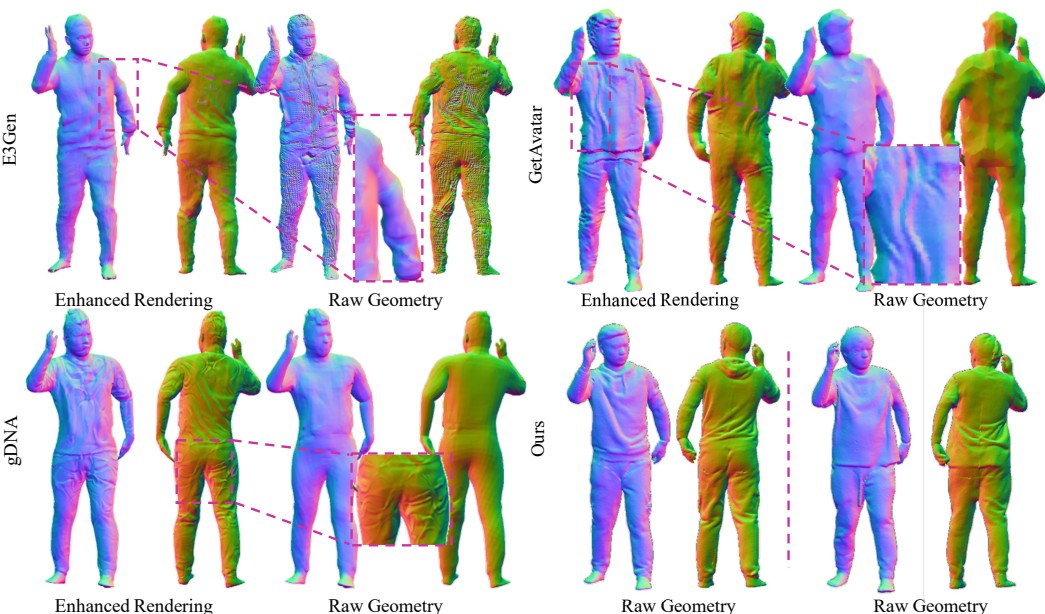

Figure 8: Qualitative comparisons of human generation results. Existing methods are displayed with both rendering results and raw geometry outputs, typically organized as two columns each. We include two geometry samples generated by our method for comparison.

field/map or predicted points' rotations. Since our goal is to compare geometry quality, we primarily evaluate these methods based on their raw geometry outputs. To ensure fairness, we also report their performance on their enhanced rendered results. For each subject, we render $50$ images from different views and synthesize $25,000$ normal images for the dataset.

We compare our method with representative human generation methods employing different 3D representations, including Gaussian splatting (E3Gen (Zhang et al., 2024d)), implicit function (GetAvatar (Feng et al., 2023), gDNA(Xu et al., 2022)), and Nerf (ENARF (Noguchi et al., 2022), GNARF (Bergman et al., 2022) and Eva3D (Hong et al., 2022)). As shown in Tab. 4, our method significantly outperforms other approaches in terms of raw geometry outputs. In comparison to the state-of-the-art (gDNA), our method improves geometry quality by $57\%$ (42.9 to 16.2). Additionally, when compared to the enhanced rendered results from other methods, our raw geometry shows improvements as well ($7\%$ from 17.4 to 16.2). As shown in Fig. 8, E3Gen (Zhang et al., 2024d) synthesizes unnatural shapes with inconsistent normal colors. GetAvatar (Zhang et al., 2023b) generates cloth wrinkles with unnatural directional patterns, while gDNA (Xu et al., 2022)'s normals produce wrinkles that appear random, unrealistic, and not coordinated with the human pose. In contrast, our method generates realistic and pose-consistent clothing details, demonstrating superior geometric fidelity and detail preservation.

## 5.4 NOVEL POSE GENERATION

Previous methods (Zhang et al., 2023b; Xu et al., 2022; Zhang et al., 2024d) synthesize 3D humans in canonical space and deform them via rigging, which limits pose-dependent clothing details (e.g., wrinkles and folds). In contrast,

Table 5: User study on quality and physical plausibility.

| Metric | GetAvatar | gDNA | E3Gen | Ours |
|---|---|---|---|---|
| Quality ↑ | 2.16 | 2.54 | 2.12 | **4.04** |
| Plausibility ↑ | 2.20 | 2.66 | 2.08 | **4.36** |

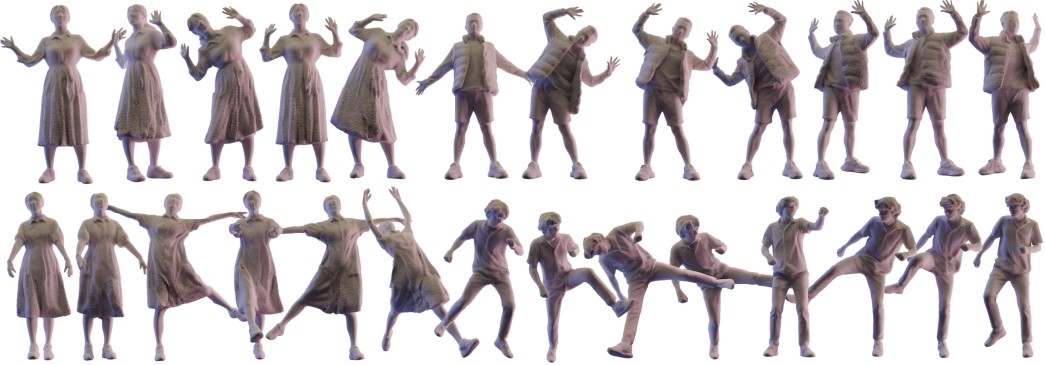

Figure 9: Generating different poses for a given avatar. We showcase challenging cases, including exaggerated poses, skirts, and loose outfits.

our approach directly synthesizes points on the
deformed human body given the pose-aware feature map, enabling the generation of pose-dependent clothing details, as shown in Fig. 9. More visual results can be found in the Appendix.

For quantitative comparisons, FID metrics reflecting geometric quality are provided in Tab. 4. Note that since prior methods' geometric details remain static across poses, their FID results correspond closely to those reported in the above table. In terms of physical plausibility of geometric details, due to the absence of standard metrics, we conduct a user study to demonstrate that our method uniquely enables pose-aware deformations that appear physically reasonable. For each method, we generate 2 identities under 8 poses and ask 25 participants to rate geometry quality and physical plausibility (1–5 scale), as shown in Tab. 5.

Our superior physical plausibility mainly stems from our synthesized pose-aware feature maps. Nevertheless, even when using a feature map that is incompatible with the body pose, our method still generates visually reasonable results, which demonstrates the robustness of our model in handling novel poses (see the Appendix for details).

## 6 LIMITATIONS

A limitation of our method lies in non-uniform sampling on the target geometry surface, as the number of target points $\mathbf{x}_1$ associated with each SMPL point $\mathbf{x}_0'$ varies. Although sufficient sampling and removing points that are too close mitigate this issue, we encourage future work to develop more advanced strategies for training pair construction. Besides, similar to other generative methods, our approach is constrained by the diversity of the training datasets. Specifically, our model generalizes well to various body types, as the dataset includes a reasonable diversity in this aspect. However, it cannot generate clothing styles that are entirely absent from the training data. Furthermore, the use of UV maps can cause seam artifacts in some randomly generated results due to discontinuity segmentation. A better approach would use UV segmentation aligned with real-world garment patches. Since our focus is on a modeling representation, addressing this issue is left for future work.

## 7 CONCLUSIONS

In conclusion, we introduce the generative human geometry distribution, the first method that integrates geometry distributions into generative modeling. This distribution-over-distribution design is novel, unique, and particularly effective, making our approach the **only one** capable of producing fine-grained geometry in a generative framework. Specifically, using flow matching, we learn a flow that transforms from the SMPL template to the target geometry, significantly improving training efficiency. Following this, our method encodes geometry into compact feature maps, facilitating various downstream generative tasks. Experimental results demonstrate that our approach achieves state-of-the-art quantitative performance compared to existing generative methods. Furthermore, our method is able to synthesize fine-grained clothing details that conform to human poses and exhibits strong robustness, generating plausible results even when provided with a feature map mismatched to the target pose. These results highlight the effectiveness of our representation and its potential for advancing 3D human modeling and synthesis.

## 8 ACKNOWLEDGEMENTS

This work was supported by funding from King Abdullah University of Science and Technology (KAUST) — Center of Excellence for Generative AI, under award number 5940.

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

# 9 APPENDIX

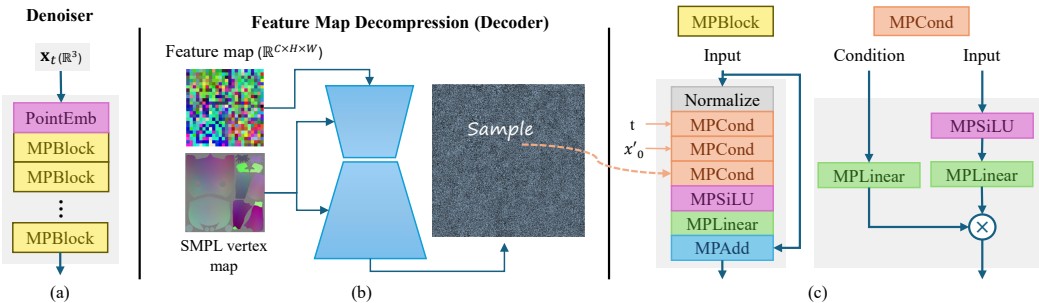

(a)            (b)            (c)

Figure 10: **Details of our auto-decoder.** (a) We sample noise from a Gaussian distribution ($\mathbb{R}^3$) and generate the flow that transforms from the SMPL template distribution to the human geometry distribution. (b) A convolutional network is used to interpret and decompress the compact feature map with the SMPL vertex map. (c) The architecture details of our model.

## 9.1 IMPLEMENTATION DETAILS

Our training requires an infinite set of training pairs $(\mathbf{x}'_0, \mathbf{x}_1)$. In practice, we sample $2^{18}$ surface points $\mathbf{x}_1$ per human geometry. We observe that many points in loose clothing regions share the same nearest SMPL template location, requiring multiple samples from that location to reconstruct these regions. To alleviate this, we sample a slightly sparser set of $2^{17}$ points on the SMPL template and apply the k-nearest neighbors (KNN) algorithm to associate each $\mathbf{x}_1$ with the nearest template point. This distributes the mapping workload across different template locations, improving the sampling efficiency for loose clothing regions. During training, we randomly sample $30,000$ pairs per iteration.

As shown in Fig. 10, the compact feature map $z_{\mathcal{T}|\mathcal{S}}$ is set to $\mathbb{R}^{8 \times 24 \times 24}$. We adopt the linear and convolutional layers from EDM2 (Karras et al., 2024; 2025), using 256 channels in all layers. For optimization, we use AdamW (Loshchilov, 2017) with a learning rate of $8 \times 10^{-3}$, following a cosine learning rate scheduler across all models. We apply L2 regularization with weight $1e^{-4}$ to the feature map during training.

We train both the auto-decoder and the generation model on the THuman2 dataset (Yu et al., 2021b) for the pose-conditioned generation task, following previous methods (Xu et al., 2022; Zhang et al., 2024d) to ensure fair comparisons. The auto-decoder model is trained for two days and the generation model for one day on a machine with two A100 GPUs. To enhance the model's ability to generalize across diverse poses, we employ the classifier-guidance that randomly drops the pose condition with a probability of $0.5$. This encourages the network to learn the underlying data distribution rather than overfitting to specific poses. For the novel pose generation task, we utilize the 4DDress dataset (Wang et al., 2024), which comprises 64 outfits across 520 human motion

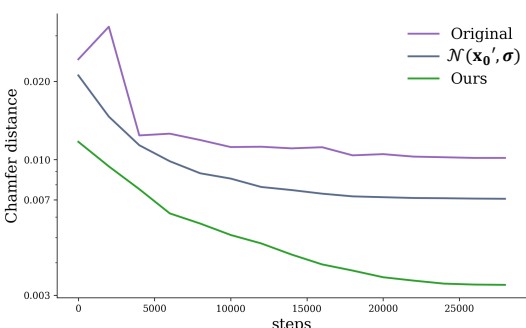

Figure 11: The evolution of the Chamfer distance of training different distribution formulations.

sequences, totaling approximately $50,000$ human geometries with diverse poses and clothing styles. We train the auto-decoder on a machine with 4 A100 GPUs for two days and train the novel pose generation model for one day. We don't utilize the classifier-guidance strategy on this dataset as its larger scale naturally supports robust training.

The auto-decoder model has 56.02M parameters. On an A100 GPU, the feature map decompression takes 81 ms, which is executed only once for the entire diffusion process. Each denoising step requires 529 ms for 1 million points, with a peak memory usage of 9522 MB. On memory-constrained

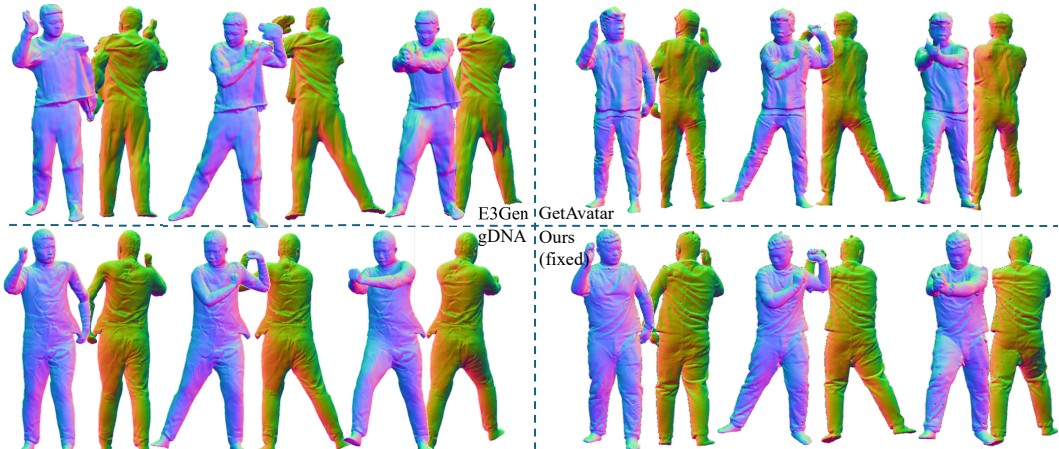

Figure 12: Demonstrations of different methods with fixed geometry features across varying poses. All results show identical geometric details in different poses.

devices, the model can sample multiple times with significantly fewer points per iteration. For instance, sampling $10,000$ points takes 71 ms with a peak memory consumption of 1420 MB.

We visualize our results by sampling 1 million points for Gaussian Splatting (GS) (Kerbl et al., 2023) rendering. Specifically, we render GS's depth maps first, from which we then extract normal maps. All points share the same scaling value, determined by the average minimum distance between points.

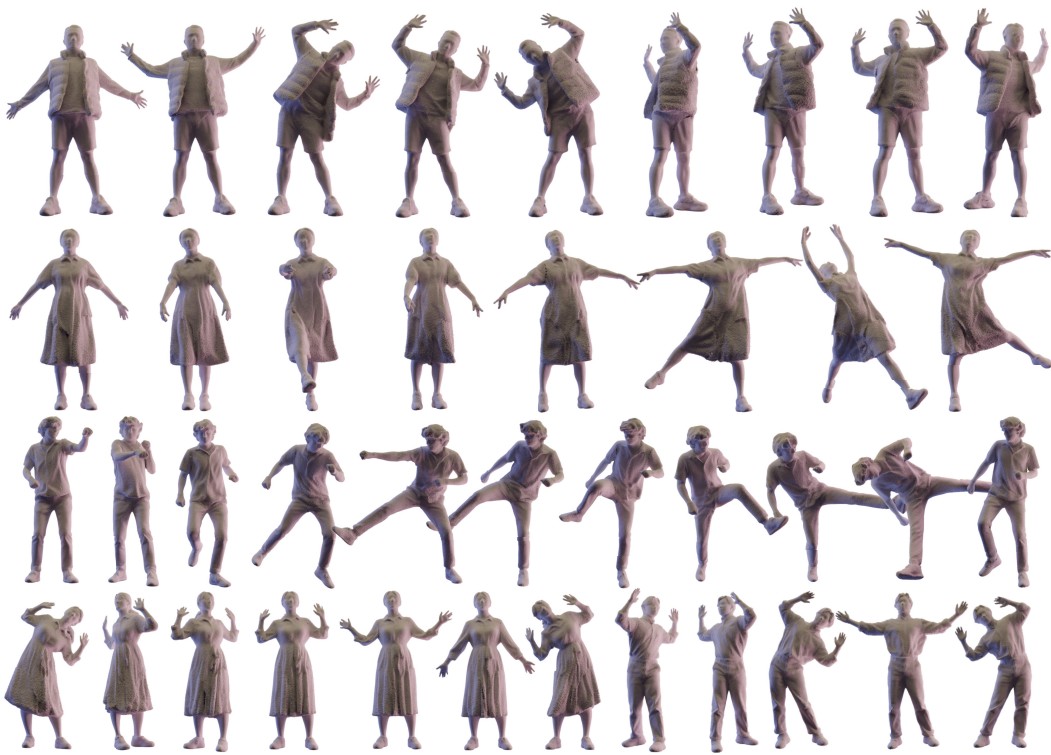

Figure 13: Novel pose generation results.

## 9.2 GEOMETRY DISTRIBUTION FORMULATION

For the experiment of different formulations in training on a dataset, the evolution of the Chamfer distance across $30,000$ iterations is illustrated in Fig. 11. Our proposed human geometry distribution achieves superior reconstruction quality, with a consistent decrease in the Chamfer distance.

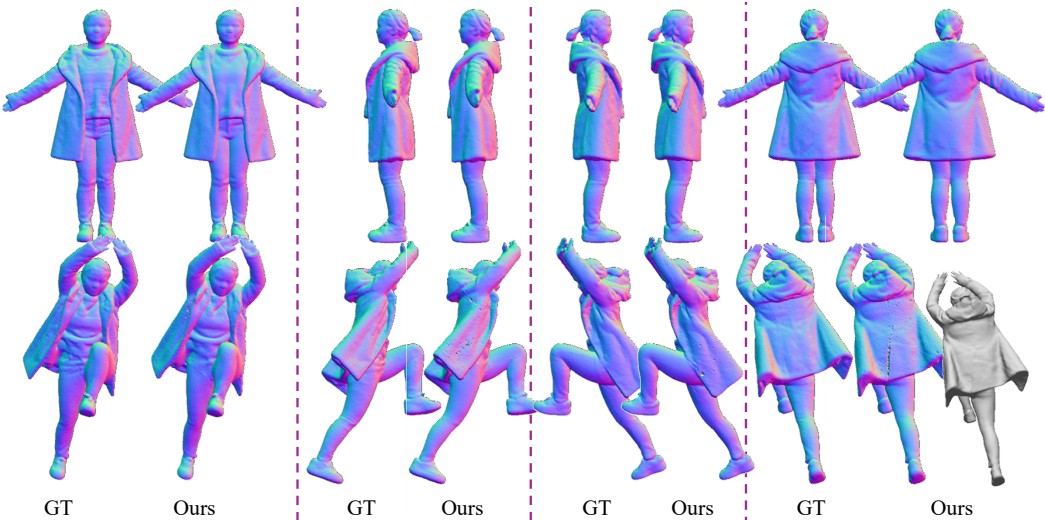

GT    Ours        GT    Ours        GT    Ours        GT    Ours

Figure 15: This figure shows challenging examples with loose clothing and hair. Our point-based representation may exhibit small local sparsity in extreme cases, but nevertheless, we maintain high geometric quality overall. To demonstrate this, we apply Poisson surface reconstruction to our results, shown on the right.

### 9.3 NOVEL POSE GENERATION

Previous methods (Zhang et al., 2023b; Xu et al., 2022; Zhang et al., 2024d) synthesize 3D humans in canonical space and obtain novel pose appearances via deformation. Their limitations are illustrated in Fig. 12, where we deform avatars to different poses, resulting in identical clothing details across all poses. In the main paper, we have showcased that our method can generate pose-aware deformations by synthesizing a pose-compatible feature map. Fig. 12 demonstrates that, although the wrinkle details may not be entirely natural, our method still produces reasonable results even when using an incompatible feature map.

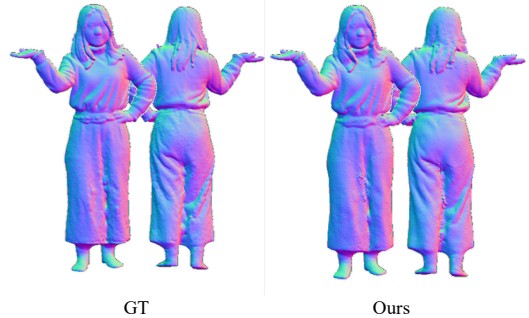

GT        Ours

Figure 14: This figure shows the results of our method after full training, using the same identities and poses as in Fig. 7.

Furthermore, we present additional novel pose generation results in Fig. 13. Specifically, we showcase challenging cases featuring intricate poses and complex clothing deformations, such as loose garments and skirts, which significantly deviate from the SMPL surface.

### 9.4 DISCUSSION ON OUR REPRESENTATION CAPABILITY

Although our representation is built upon the SMPL template, it differs fundamentally from prior methods whose generated geometries are strictly confined by SMPL. In contrast, our representation models the actual target geometry distribution, without being limited by the SMPL template. As illustrated in Fig. 15 and Fig. 16, our method successfully captures loose clothing and hair—details that cannot be defined or represented by the SMPL model.

Our representation naturally generalizes to different body shapes and genders, as it models the distribution of displacements between the SMPL template and the target geometry, as illustrated in Fig. 17.

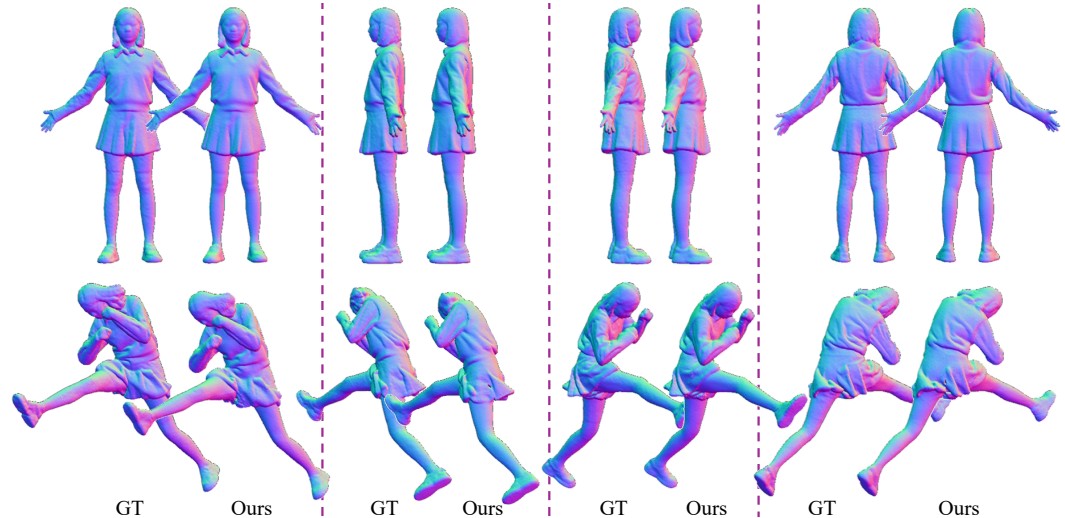

Figure 16: This figure illustrates large deformations of loose clothing under extreme poses.

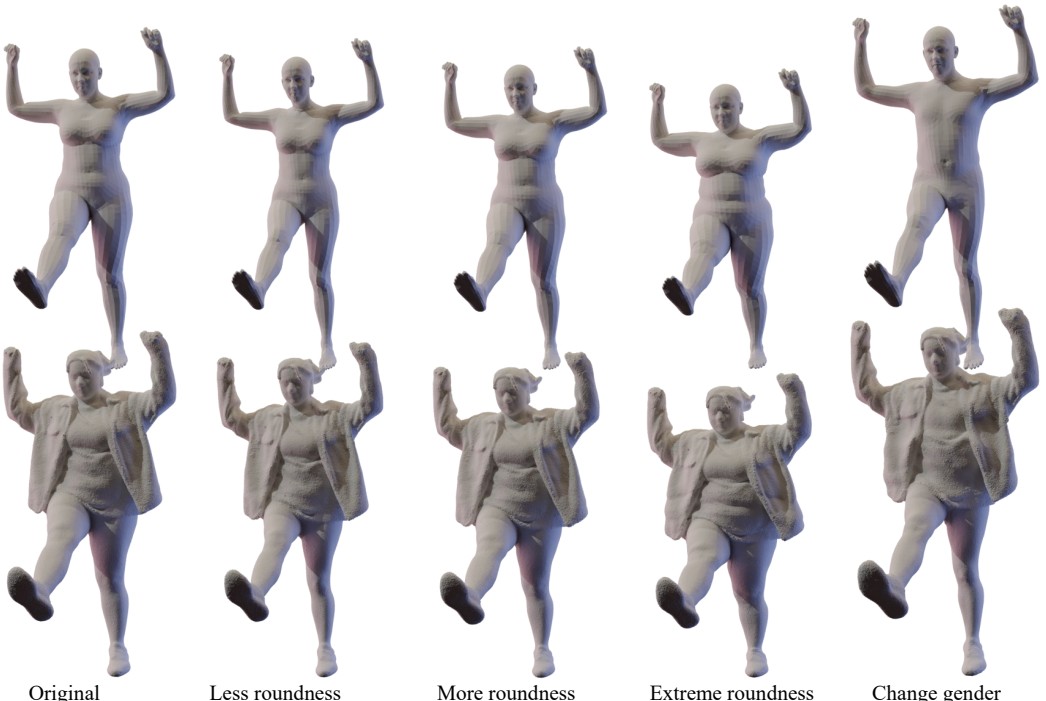

Figure 17: Given the same feature map, the figure shows the generated results for different SMPL body shapes and genders. Since the underlying identity feature map remains unchanged, the gender differences mainly appear as subtle variations in body shape.

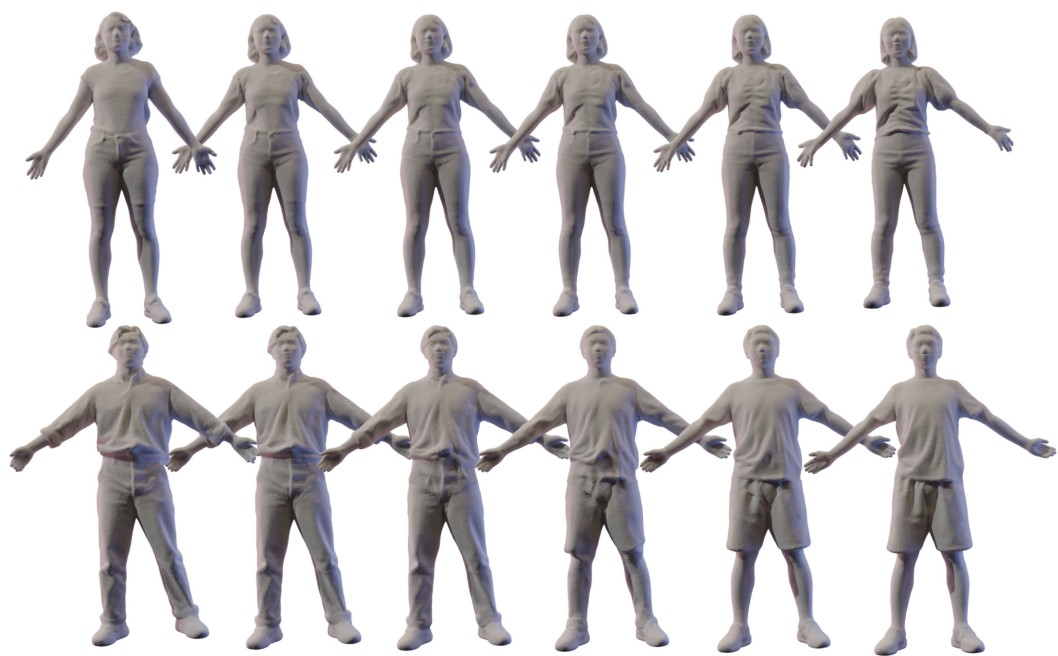

Figure 18: The figure shows interpolation results between the leftmost and rightmost identities; note how new clothing wrinkles, styles, and hairstyles naturally emerge during the transition.

## 9.5 IDENTITIES/GARMENTS INTERPOLATION

In addition to training another generative network, our method also enables the generation of novel clothing and identities through latent-space interpolation, as shown in Fig 18 and Fig. 19.

## 9.6 USE OF LARGE LANGUAGE MODELS

During the preparation of this manuscript, we employed a large language model (LLM) as an auxiliary tool to assist with grammar checking and minor language refinements. All scientific content, ideas, and claims were conceived and written by the authors. The final version of the manuscript was carefully reviewed, modified, and approved by the authors to ensure accuracy and clarity.

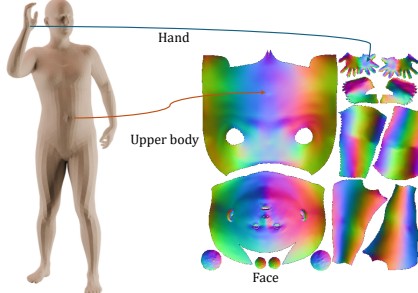

Figure 20: UV parameterization of the SMPL template.

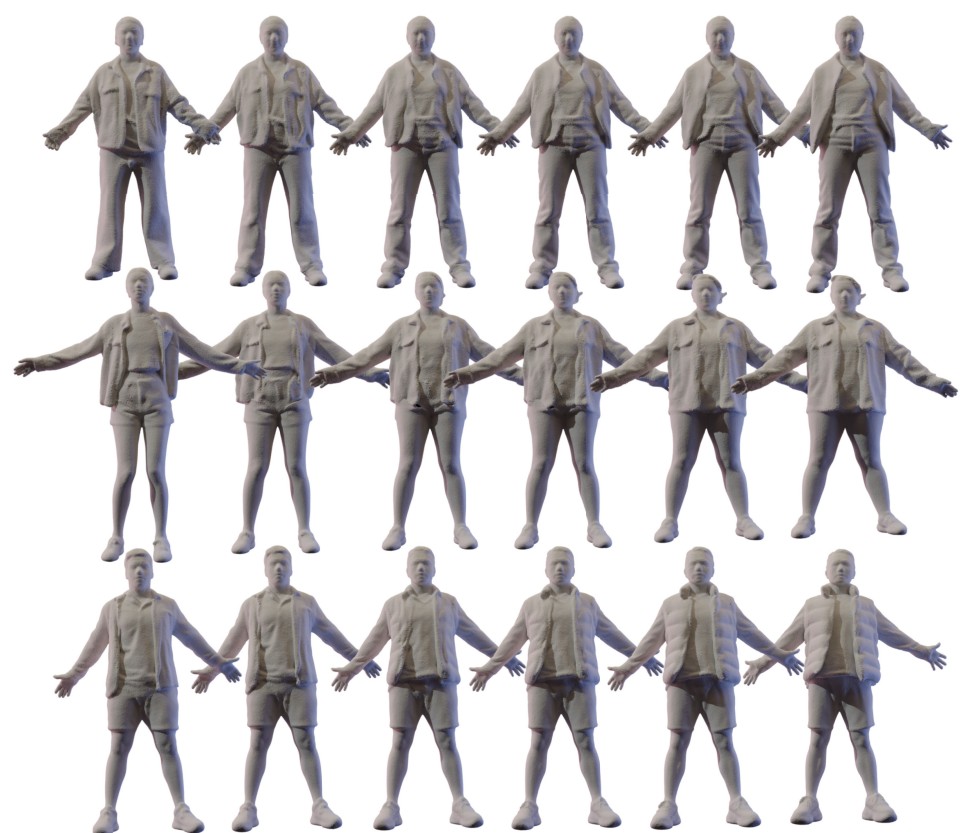

Figure 19: Interpolations. This figure presents interpolations between identities wearing outerwear.

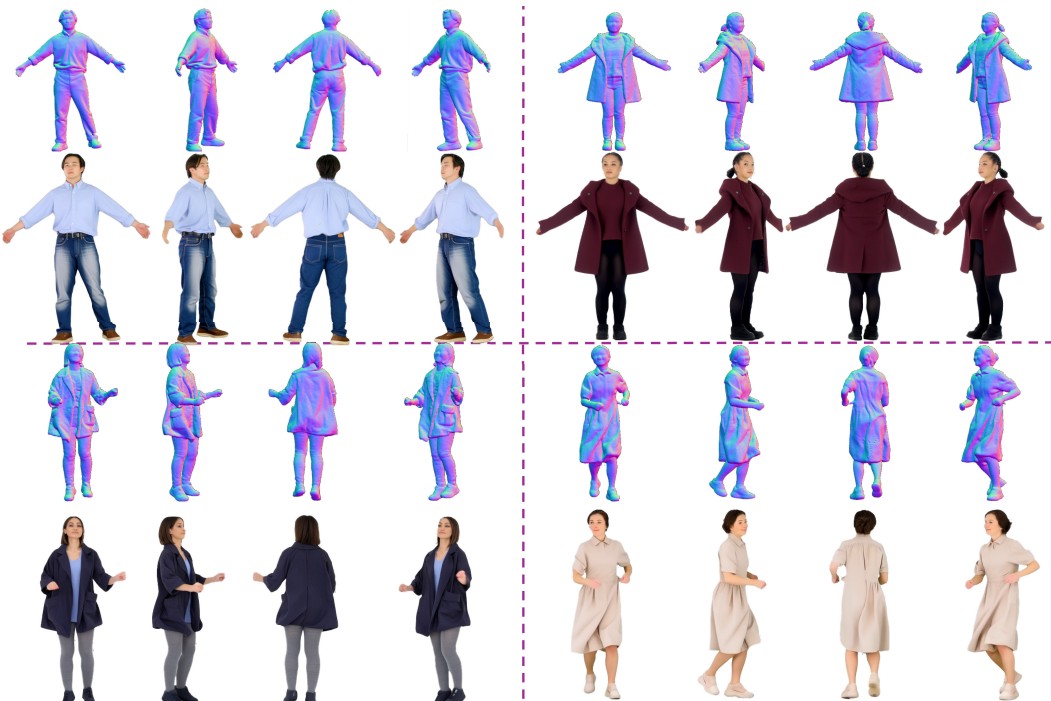

Figure 21: Given our high-quality geometries, textures can be synthesized in various ways. Here, we first generate multi-view images with a depth-conditioned video generation model (Wan2.1), and then optimize point colors via Gaussian Splatting.

