# OpenReview forum: "Generative Human Geometry Distribution"
_ICLR.cc/2026/Conference — ICLR 2026 Oral_

### Official Review · Reviewer_aJPB · 2025-10-17

**Soundness:** 3
**Presentation:** 3
**Contribution:** 3
**Rating:** 6
**Confidence:** 4

**Summary:**

This paper addresses the challenge of 3D human geometry generation by proposing a new geometry distribution model that extends this representation from single characters to large-scale dataset learning.

Specifically, its proposed representation incorporates two key designs:
(1) Encode geometry distributions as 2D feature maps instead of network parameters to enable dataset-level generalization.
(2) Replace the Gaussian source distribution with the SMPL human template (plus refined flow velocity fields) to align the source closer to the target geometry distribution.

Based on this representation, this paper also proposes a two-stage generative framework:
(1) Compress each human geometry distribution into a compact 2D feature map using a flow model, from which a high-fidelity human geometry can be sampled through a denoising process.
(2) Train a second flow model on the latent feature space to support generative tasks.

The method is evaluated on two tasks: pose-conditioned random avatar generation (THuman2 dataset) and avatar-consistent novel pose synthesis (4DDress dataset). Quantitative results show a 57% improvement in geometry quality and a 7% improvement in visual appearance compared to SOTA.

**Strengths:**

1. The technical designs in the proposed geometry distribution are solid:
(1) Replacing Gaussian with SMPL (a human-specific template) reduces the "distance" between source and target distributions, minimizing the need to learn extraneous flow paths. This, combined with training pair construction (matching target points to nearest SMPL points + Gaussian perturbation) and distribution normalization (zero-centered Gaussian for source, dense displacement fields for target), solves spatial imbalance in training and accelerates convergence.
(2) Unlike prior geometry distributions that store shape information in network weights, encoding into 2D feature maps enables efficient dataset-level learning. Plus, the idea of using the decompressed feature map to serve as a condition to generate the final geometry distribution makes sense to me.

2. The quantitative and qualitative experiments are comprehensive, showcasing the effectiveness of the proposed method as well as each design component.

**Weaknesses:**

Please refer to the question section.

**Questions:**

1. I'm kind of uncertain about the application scenario. This method is built upon you already have the SMPL parameters, right? If SMPL can already provide human geometry information, what's the role of this proposed method? Is it more like a refiner?

2. I'm a little confused about the decoding process. Looks like you're optimizing the flow model $\theta$ and the condition (i.e. the 2D feature map $\textbf{z}_{T|S}$) simultaneously. Will the training process be stable since I recall that normally in diffusion training, the condition should be fixed, right?

3. I'm wondering do you have a mathematical analysis about why recovering target geometry by conditioning on SMPL can be modeled as a 2D feature map? What does each pixel in the feature map mean? Maybe the author should give more introduction or preliminary knowledge on the 2D feature map representation of 3D objects like SMPL.

4. I'm curious about the influence of feature map size. The paper fixes the feature map to 8×24×24. Are 576 embeddings too few to encode 50,000 points?

---

> ### Author Response · Authors · 2025-11-19
>
> We sincerely thank the reviewer for the thoughtful and positive comments. We provide detailed responses to all questions and suggestions below. In addition, we list here the newly added figures in the Appendix for clarity.
>
> Fig. 15 and Fig. 16: Examples of fine-grained details beyond the SMPL template, including loose clothing and hair.
>
> Fig. 20: Illustration of 2D UV parameterization of 3D geometry.
>
> **Application scenario of the SMPL is given. If SMPL can already provide human geometry information, what's the role of this proposed method?**
>
> The SMPL template only represents coarse human shapes and cannot capture clothing, hair, identity-specific features, or other fine-grained details. Our method generates them (See the newly added Fig. 15 and Fig. 16 in the Appendix) using the SMPL model to parametrize the distribution better. Besides, the SMPL template is easy to obtain, so using it as a prior is common in human generation methods (e.g., PrimDiffusion[1], HQ-Avatar[2], XAGen[3], StructLDM[4]).
>
> **Will the training process of auto-decoder be stable if optimizing the flow model and the condition simultaneously**
>
> We follow the standard auto-decoder framework and simultaneously optimize both the latent codes and network parameters, similar to DeepSDF[5]. In practice, using our diffusion backbone, we did not observe any stability issues.
>
> **Why can human geometry be modeled as a 2D feature map?**
>
> The SMPL template provides a 2D UV parameterization of 3D geometry, where each pixel of the vertex map represents a 3D point on the SMPL surface. We add Fig. 20 in the Appendix for illustration. Since each geometric point in our method paris with an SMPL point, we can use SMPL's 2D parameterization to represent humans. Typically, the information in a texel in the feature map tells the flow model how to transport a point from the surface of the SMPL model to a closeby point on the surface of the final human geometry. However, clothing details can be reasonably far away from the surface of the SMPL model. That means, geometric details can be closeby or in medium distance from a texel in the 2D feature map.
>
> **What does each pixel in the feature map mean? Are 576 embeddings too few to encode 50,000 points?**
>
> The 8×24×24 feature map is a compressed representation, with each pixel roughly encoding geometric features of its corresponding 3D position. Because it is a compressed latent, it captures correlations across points and thus can represent far more information than the 576 pixels alone. This concept is similar to latent auto-encoders [6], which also provided significant detail using only a small-resolution compressed representation.
>
> It is therefore true that  the seemingly low resolution feature map contains enough information to provide geometric information for “50,000 points”. In fact, the 2D map is a continuous representation, which allows effectively infinite sampling and is not limited by a fixed number of points. We can generate meaningful geometric information for more than 50K points and show results for up to 1M points in the paper. Of course we make use of the fact that we are specializing a human. If we would add other geometry, such as trees, cars, and fish to the dataset, they would not be parametrized well by the SMPL model and the 2D feature map.
>
> [1] Chen, Zhaoxi, et al. "Primdiffusion: Volumetric primitives diffusion for 3d human generation." Advances in Neural Information Processing Systems 36 (2023): 13664-13677.
>
> [2] Zhang, Weitian, et al. "HQ-Avatar: Towards High-Quality 3D Avatar Generation via Point-based Representation." 2024 IEEE International Conference on Multimedia and Expo (ICME). IEEE, 2024.
>
> [3] Xu, Zhongcong, et al. "Xagen: 3d expressive human avatars generation." Advances in Neural Information Processing Systems 36 (2023): 34852-34865.
>
> [4] Hu, Tao, Fangzhou Hong, and Ziwei Liu. "Structldm: Structured latent diffusion for 3d human generation." European Conference on Computer Vision. Cham: Springer Nature Switzerland, 2024.
>
> [5] Park, Jeong Joon, et al. "Deepsdf: Learning continuous signed distance functions for shape representation." Proceedings of the IEEE/CVF conference on computer vision and pattern recognition. 2019.
>
> [6] Rombach, Robin, et al. "High-resolution image synthesis with latent diffusion models." Proceedings of the IEEE/CVF conference on computer vision and pattern recognition. 2022.

---

### Official Review · Reviewer_2RbH · 2025-10-25

**Soundness:** 3
**Presentation:** 3
**Contribution:** 3
**Rating:** 6
**Confidence:** 4

**Summary:**

This paper has proposed a novel 3D human generative model built on top of Geometric Distribution. Specifically, it learns a conditional flow model from the SMPL parameter to the loose cloth real geometric distribution. Some training tricks (pair sampling, distribution normalization, and auto-decoder) are proposed to improve the performance. Extensive experiments have demonstrated the effectiveness of the proposed method.

**Strengths:**

1. The method is novel, and the problem is worth exploring.
2. The proposed training tricks are reasonable and effective.
3. The final quality is surprisingly good.

**Weaknesses:**

1. Since some baselines (Like EVA3D) also model the texture distribution, directly comparing the geometry with it is somewhat unfair. But this is not a big issue.
2. Some of the proposed training tricks are actually already well established in the previous works. E.g., the SMPL -> final distribution idea is similar to the diffusion bridge, and the DistNorm is very popular in the current diffusion model training (vae rescaling).
3. The proposed method has not demonstrated the performance on the animation / temporal consistency.

**Questions:**

1. If autodecoder is used, when new data comes, does it mean that this proposed method cannot handle this case? Or it requires some test-time optimization, which is unscalable.
2. The UV-space diffusion + autodecoder pipeline for 3d avatar generation is very relevant to Gaussian3Diff (ECCV 24), and should be discussed in the related work.

---

> ### Author Response · Authors · 2025-11-19
>
> Thank you for the positive comments of our results and appreciate the constructive feedback. We provide detailed responses to the questions and suggestions below.
>
> **The performance on the animation / temporal consistency**
>
> The task does not target temporal consistency, as each frame is generated independently. Its purpose is to show that our method can produce plausible wrinkles and clothing deformations across different poses.
>
> **Does using an autodecoder mean the method cannot handle new data?**
>
> Handling new data is challenging for the autodecoder. Normally, we need test-time optimization or another network (e.g. our proposed generative network) to get a latent for a novel avatar. However, the autodecoder is a pre-process of training the generative model, and any autoencoder also has to be retrained if new data becomes available.
>
> **Related work: Gaussian3Diff**
>
> Thank you for the comments. We have added it in the related work.

---

### Official Review · Reviewer_epmJ · 2025-11-01

**Soundness:** 3
**Presentation:** 3
**Contribution:** 3
**Rating:** 8
**Confidence:** 3

**Summary:**

Building upon the method Geometry Distribution, this paper introduces a human generation method via a flow-based diffusion model. To enable efficient training over the existing human datasets, the authors first propose to encode the human geometry distribution into a feature map. Leveraging the pre-trained feature map, the authors further adopt the SMPL model distribution as the learning base while refining the flow velocity field for more efficient training. The paper also introduce an efficient strategy for constructing training pairs, further improving the generated quality. Extensive experiments have been conducted to demonstrate the performance of the proposed method across various tasks.

**Strengths:**

The strengths of the method can be summarized as:

+ The adoption of geometry distributions and the introduction of feature map "representation" is novel and demonstrated to be useful.

+ Both the qualitative and quantitative evaluations are thorough and demonstrate the performance of the proposed method, establishing a new state-of-the-art.

+ The paper is well-written and easy to follow.

**Weaknesses:**

The reviewer would like to point out several weaknesses of the paper:

- The method may still inherit some limitations from the SMPL template, such as the modeling of fine details like hair, accessories, hats, etc.

- Based on the first weakness, the reviewer would like to see the rotating results of loose clothing, complex poses.

- Considering the method is trained on THUman2 dataset, it might be beneficial to compare with the 3D reconstruction methods (like PIFu, ECON, ICON) in terms of both qualitative and quantitative quality to further demonstrate the method or present the gap. Additionally, more metrics like P2S and chamfer distance may be useful to help demonstrate the 3D quality.

**Questions:**

1. In Figure 7, we could still observe some artifacts in "Ours", especially for the frontal view. However, in Figure 6, it's presented that the proposed strategy for constructing training pairs could address (or actually minimize) this issue. THe reviewer would like to ask the potential and actual factors that lead to this problem, and also the possible solutions.

2. Considering the method employs the SMPL model as the base, will that be possible to generate different shapes (tall, short, fat, thin) or even genders for a human subject?

---

> ### Author Response · Authors · 2025-11-19
>
> We sincerely thank the reviewer for the thoughtful and positive comments. We provide detailed responses to the questions and suggestions below. In addition, we list here the newly added figures in the Appendix for clarity.
>
> Figure 14:  Results of our method after full training, using the same identities as in Fig. 7.
>
> Figure 15 and 16: Rotating results showing fine-grained details beyond the SMPL template, including loose clothing and hair.
>
> Figure 17: Results for different SMPL body shapes and genders.
>
> **Method may inherit some limitations from the SMPL template. Rotating results of loose clothing, complex poses.**
>
> Our method is capable of generating geometric details that go beyond the representational limits of the SMPL template. Newly added rotation results of characters in extreme poses, showing loose clothing and hair, are provided in the appendix (Fig 15, 16).
>
> **Comparison with 3D reconstruction methods.**
>
> We thank the reviewer for the suggestion. Our work focuses on generative tasks, where ground-truth geometries may not exist and exact matching is not the goal, which differs from reconstruction methods.  To provide a reference, we report the Chamfer distance of the ECON reconstruction (3.1e-3) and that of our latent reconstruction (2.3e-6) for 50 geometries in Dress4d dataset.
>
> **Artifacts in Figure 7**
>
> The results shown in Figure 7 were obtained before full convergence. We add the fully trained results in the appendix for reference (Fig. 14).
>
> **Is it possible to generate different shapes (tall, short, fat, thin) or even genders?**
>
> Yes, our method can naturally extend to different shapes and genders (see the newly added Fig. 17 in the Appendix).

---

### Official Review · Reviewer_P4Gu · 2025-11-01

**Soundness:** 2
**Presentation:** 2
**Contribution:** 2
**Rating:** 2
**Confidence:** 5

**Summary:**

This paper proposes to learn the generative human geometry distribution from 3D human dataset, which can further be applied for random human generation and animation. At the core of the method is a diffusion model which is conditioned on the UV space of SMPL model to predict the geometry distribution. The authors conducted experiments on different dataset to verify the effectiveness of the proposed method.

**Strengths:**

The paper is easy to follow.

Generating 3D humans is an interesting task with practical applications.

The method is technically sound by leveraging a UV-conditioned diffusion model to learn the geometry distribution of humans.

**Weaknesses:**

Insufficient experiments. The paper only evaluates the method on studio 3D data, and does not illustrate whether the proposed method can generate out-of-domain 3D humans. The paper does not answer the question of whether the method overfits to the training dataset, and whether it can sample humans with novel clothing styles. Only random sampling and novel pose generation are shown.

In addition, in Fig. 4, the authors claim that the pipeline supports multiple condition,s such as images/texts, which, however, are not illustrated in the paper. Can the method generate 3D humans given single in-the-wild images, or generate text-conditioned humans like joint2human?


Existing methods can learn 3D human generations from 2D data such as images and videos, including GAN-based (e.g., EVA) and diffusion-based method (e.g., Primitive Diffusion [Zhao et al. NeurIPS, StructLDM), or generate both geometry and textures using 3D Gaussians in a feed-forward way, such as LHM, AniGS. Compared with these work, what are the advantages of the proposed method which require 3D ground truth data in training and can only generate 3D geometry?

This paper (video demo) claims that it is the first method that learn the human geometry distribution of a dataset. However, this statement is not true. Previous 3D human generation methods such as GDNA [Xu et al 2022] for geometry reconstruction, or EVA/PrimDiff for both geometry and texture reconstruction all learn the distribution of human geometry.

**Questions:**

The method utilizes SMPL model. How does this handle loose clothing?

---

> ### Author Response · Authors · 2025-11-19
>
> We sincerely thank the reviewer for the thoughtful and detailed comments. We first clarify concerns that seem to arise from a misunderstanding of our method, and then address the remaining points individually.
>
> We list here the newly added figures in the Appendix for clarity.
>
> Fig. 18: Interpolation between two different latents
>
> Fig. 19: Interpolation between two different latents
>
> Fig. 21: Texture synthesis result for our geometry
>
> **Previous method also learns the dataset distribution.**
>
> We would like to clarify that our method does not merely model the dataset distribution; rather, it models the dataset distribution of individual geometry distributions. The key distinction from previous approaches is that we treat each human geometry itself as a distribution. This is the core contribution of our work, resulting in significantly higher geometric quality and finer details in our results.
>
> **What are the advantages of the proposed method, considering that some methods only require 2D data and can generate texture?**
>
> The core contribution of our method is to produce 3D geometry with detail and quality approaching real-world scans (outperforming the baseline by 57%). The difference in quality is visually apparent, as shown in our figures and supplementary videos. Besides, using 3D data for training is a promising and increasingly common strategy in state-of-the-art 3D generation (e.g. Hunyuan3D [1], TRELLIS [2], Direct3D [3]). While learning 3D human geometry from images is a valid and important topic, more recent generative methods focus on explicitly training on geometry. Both strategies should be researched.
>
> Once high-quality geometry is available, textures can be obtained using existing texture-synthesis techniques, making geometry the core challenge in this problem. We provide textured examples in the Appendix Fig. 21.
>
> **On supporting other conditioning modalities**
>
> Our framework is inherently modality-agnostic, provided suitable training data are available. We validate this using two tasks with different conditioning in the paper (SMPL and image conditions),  and in principle, the image embedding could be replaced with a text embedding to enable text-conditioned generation.
>
> **On overfitting and out-of-domain generalization**
>
> Our latent model does not merely memorize the dataset; instead, it learns a smooth latent space that supports meaningful generalization. For example, interpolations between latents produce plausible new variations beyond those seen in the dataset (we added new examples in the Appendix Fig. 18 and Fig. 19).
>
> Due to the limitations of current 3D datasets, downstream models for generative tasks may reproduce clothing styles present in the dataset, a phenomenon also observed in prior works using the same dataset (e.g. GetAvatar [4], E3Gen [5], HQ-Avatar [6]). Therefore, this line of research primarily focuses on generating high-quality and diverse geometries under the same pose, rather than synthesizing entirely new garment categories (Fig. 8 demonstrates such intra-pose diversity).
>
> **The method utilizes SMPL model. How does this handle loose clothing?**
>
> We use SMPL for parametrizing a distribution and not as final geometry. The distribution-based representation can model arbitrary geometry with any topology or shape, so the generated geometry can significantly deviate from SMPL models to produce loose garments.
>
> [1] Zhao, Zibo, et al. "Hunyuan3d 2.0: Scaling diffusion models for high resolution textured 3d assets generation." arXiv preprint arXiv:2501.12202 (2025).
>
> [2] Xiang, Jianfeng, et al. "Structured 3d latents for scalable and versatile 3d generation." Proceedings of the Computer Vision and Pattern Recognition Conference. 2025.
>
> [3] Wu, Shuang, et al. "Direct3d: Scalable image-to-3d generation via 3d latent diffusion transformer." Advances in Neural Information Processing Systems 37 (2024): 121859-121881.
>
> [4] Zhang, Xuanmeng, et al. "Getavatar: Generative textured meshes for animatable human avatars." Proceedings of the IEEE/CVF International Conference on Computer Vision. 2023.
>
> [5] Zhang, Weitian, et al. "E3Gen: Efficient, Expressive and Editable Avatars Generation." Proceedings of the 32nd ACM International Conference on Multimedia. 2024.
>
> [6] Zhang, Weitian, et al. "HQ-Avatar: Towards High-Quality 3D Avatar Generation via Point-based Representation." 2024 IEEE International Conference on Multimedia and Expo (ICME). IEEE, 2024.

---

### Meta-Review · Area_Chair_o9z2 · 2025-12-29

**Summary:**

The only outlier review is from Reviewer *P4Gu* (2 -- 866). The AC believes Reviewer *P4Gu* misunderstands the main goal of this work: to learn the "distribution of distributions" from 3D clothed human data. The first distribution pertains to dataset distributions, while the second models human shape, allowing points to be sampled at arbitrary resolutions. A 3D-native generative model offers advantages and typically produces more detailed shapes than 2D solutions like SDS-based or multi-view methods. AC also agrees with Reviewer *P4Gu*'s suggestion to remove the "text" label from Figure 4 or consider adding a new figure to showcase its potential in text-to-human tasks.

Overall, since the main concerns of the other three reviewers have been addressed, and the AC believes Reviewer P4Gu would increase the score to 4 or even 6 with full engagement, the AC recommends accepting this paper. It also encourages the authors to incorporate all feedback into the final version.

This novel approach to modeling clothed humans as SMPL-conditioned point distributions could lead to future advancements in lossless shape compression, shape matching, and body fitting, warranting an **Oral** presentation.

**Reviewer Concerns:**

- Reviewer P4Gu: concerns are well addressed
- Reviewer epmJ: concerns are well addressed
- Reviewer 2RbH: all concerns are well explained but the limited novelty (diffusion bridge + vae rescaling)
- Reviewer aJPB: concerns are well addressed

**Reviewer Scores:**

- Reviewer P4Gu should raise the score (4 $\rightarrow$ 6)
- Reviewer epmJ should keep or even raise the score (8 $\uparrow$)
- Reviewer 2RbH should keep or might lower the score (6 $\downarrow$)
- Reviewer aJPB should keep or even raise the score (6 $\uparrow$)

---

### Decision · Program_Chairs · 2026-01-26

Accept (Oral)